# Metabolic Dependency Shapes Bivalent Antiviral Response in Host Cells in Response to Poly:IC: The Role of Glutamine

**DOI:** 10.3390/v16091391

**Published:** 2024-08-30

**Authors:** Grégorie Lebeau, Aurélie Paulo-Ramos, Mathilde Hoareau, Daed El Safadi, Olivier Meilhac, Pascale Krejbich-Trotot, Marjolaine Roche, Wildriss Viranaicken

**Affiliations:** 1PIMIT—Processus Infectieux en Milieu Insulaire Tropical, INSERM UMR 1187, CNRS 9192, IRD 249, Plateforme CYROI, Université de La Réunion, 97490 Sainte-Clotilde, France; 2Diabète Athérothrombose Réunion Océan Indien (DéTROI), INSERM UMR 1188, Campus Santé de Terre Sainte, Université de La Réunion, 97410 Saint-Pierre, France

**Keywords:** antiviral response, OXPHOS, mitochondrial respiration, glycolysis, metabolic reprogramming, immunometabolism

## Abstract

The establishment of effective antiviral responses within host cells is intricately related to their metabolic status, shedding light on immunometabolism. In this study, we investigated the hypothesis that cellular reliance on glutamine metabolism contributes to the development of a potent antiviral response. We evaluated the antiviral response in the presence or absence of L-glutamine in the culture medium, revealing a bivalent response hinging on cellular metabolism. While certain interferon-stimulated genes (ISGs) exhibited higher expression in an oxidative phosphorylation (OXPHOS)-dependent manner, others were surprisingly upregulated in a glycolytic-dependent manner. This metabolic dichotomy was influenced in part by variations in interferon-β (IFN-β) expression. We initially demonstrated that the presence of L-glutamine induced an enhancement of OXPHOS in A549 cells. Furthermore, in cells either stimulated by poly:IC or infected with dengue virus and Zika virus, a marked increase in ISGs expression was observed in a dose-dependent manner with L-glutamine supplementation. Interestingly, our findings unveiled a metabolic dependency in the expression of specific ISGs. In particular, genes such as ISG54, ISG12 and ISG15 exhibited heightened expression in cells cultured with L-glutamine, corresponding to higher OXPHOS rates and IFN-β signaling. Conversely, the expression of viperin and 2′-5′-oligoadenylate synthetase 1 was inversely related to L-glutamine concentration, suggesting a glycolysis-dependent regulation, confirmed by inhibition experiments. This study highlights the intricate interplay between cellular metabolism, especially glutaminergic and glycolytic, and the establishment of the canonical antiviral response characterized by the expression of antiviral effectors, potentially paving the way for novel strategies to modulate antiviral responses through metabolic interventions.

## 1. Introduction

Over the past two decades, we have been regularly confronted with our vulnerability to infectious diseases, whether emerging or re-emerging [1,2,3,4,5,6,7]. To deal with epidemics that can take on worldwide proportions, major research efforts have been required to gain a better understanding, not only of the emergence of these viruses, but also the pathophysiology associated with them. The antiviral response is one of the most widely explored aspects, depicting the host–pathogen relationship that develops during infection, which represents an evolutionary mechanism elaborated by the host to struggle against viral infections [8]. The canonical pathway of the antiviral response—the interferon (IFN) pathway—is initiated by the recognition of pattern-associated molecular patterns (PAMPs), usually viral nucleic acids, by pattern recognition receptors (PRRs). The viral genome released into the cytosol is specifically recognized by RIG-I like receptors (RLR; e.g., RIG-I and MDA5) notably for viral 5′PPP-RNA and cyclic GMP-AMP Synthase (cGAS) for viral dsDNA [9,10]. Nucleic acids of endosomal origin are recognized by toll-like receptors (TLRs) found in the endosome; TLR3 for dsRNA, TLR7/8 for ssRNA and TLR9 for DNA containing unmethylated CpGs [9]. Most of these receptors activate interferon regulatory factors (IRFs) via adaptor proteins. For example, RIG-I and MDA5 interact with MAVS via their CARD domain, leading to a signaling cascade involving TRAF3, TBK1 and IKKε, resulting in phosphorylation and heterodimerization of IRF7 and IRF3 [9]. Similarly, TLR3, via TRIF, leads to the activation of IRF3 and 7 while TLR7, 8 and 9, via MYD88, lead to the formation of IRF7 homodimers [9]. More recently, the demonstration of the ability of cGAS to produce cyclic GMP-AMP (cGAMP) following the detection of cytosolic dsDNA led to the description of a new IFN production pathway via the Stimulator of Interferon Genes—STING [10]. STING, through TBK1, leads to the phosphorylation and homodimerization of IRF3 [10]. Finally, IRFs activated by these different pathways can translocate to the nucleus of the infected cell, where they interact with their promoter to allow the transcription of type I and III IFNs. It is difficult to distinguish the intrinsic activity of type I IFNs (predominantly β) from type III IFNs, as their secretion occurs concomitantly [11]. Secreted IFNs act in an autocrine and paracrine fashion, binding to their respective IFNAR1/2 and IFNLR1/IL-10Rβ receptors. This triggers the phosphorylation and homodimerization of STAT1 (GAF) transcription factor. It also leads to the formation of the ISGF3 complex, comprising IRF9 and phosphorylated STAT1 and STAT2 proteins [11,12]. The ISGF3 complex activates transcription by binding to interferon-sensitive response elements (ISREs), while the GAF complex does so by targeting gamma interferon activation sites (GASs); both mechanisms culminate in the upregulation of interferon-stimulated genes (ISGs) [11,12]. Alternatively, specific antiviral programs have been demonstrated, involving, for example, peroxisomal localization of the MAVS which, after interaction with RIG-I, specifically induces activation of IRF1 leading to the direct expression of antiviral genes and, at the epithelial level, IFN-λ [13,14]. The ISGs encode a range of antiviral proteins that exhibit diverse functions throughout the viral life cycle. These functions include disrupting viral entry (myxovirus resistance genes), impeding viral translation and replication (IFIT family, viperin), and even interfering with viral budding (tetherin) [11]. Furthermore, these genes have the capacity to enhance the detection of pathogens and amplify interferon signaling cascades [15].

The IFN response is acute, representing the first line of defense deployed by cells at the site of infection, limiting the spread of the virus throughout the body. It also enables the innate and adaptive immune responses to be established accordingly [16]. However, this line of defense is vulnerable and many viruses have developed mechanisms to circumvent and control the antiviral response. To avoid the IFN response, several of these mechanisms are often active simultaneously. Upstream of IFN secretion, certain viruses appear to act by limiting the detection of their genome or by interfering with the signaling pathways leading to IFN secretion. For example, the N and NSP5 proteins of SARS-CoV-2 limit the polyubiquitination of RIG-I, which is involved in the stabilization of its CARD domains and the aggregation of MAVS [17]. Viral protease from HCV has shown the ability to cleave Cardif, an adaptor protein in the RIG-I antiviral pathway [18]. Viral escape can also occur downstream of IFN secretion by inhibiting ISG production. Many viruses (e.g., Nipah virus, Measles virus, DENV, ZIKV) are able to target STAT1 and STAT2, sequestering them or causing their degradation [19]. Furthermore, while metabolism has proven to be of interest in the activation of immune cells [20], it is no less central to the IFN response. Metabolism is a crucial control point for the virus during infection in several regards. Firstly, during infection, the diversion of cellular metabolism from OXPHOS to glycolysis, via a Warburg-like effect, allows the virus to access a quantity of biomass consistent with viral progeny [21]. On the other hand, control of metabolism could limit its effects on the establishment of the antiviral response. Indeed, we have previously shown that induction of the antiviral response by poly:IC in galactose-cultured cells, to boost their OXPHOS, resulted in greater expression of ISGs than when cells based their metabolism on glucose use and glycolysis [22]. Consistent with our results, mitochondrial ROS production was previously shown to induce MAVS oligomerization and type I IFN production [23]. In support of this hypothesis, it was shown that cells resistant to HIV-induced aerobic glycolysis have a higher antiviral potential and are accompanied by virus control [24]. Strengthening the link between cell respiration and virus infection, flavivirus affects the respiratory pathways of infected cells, notably through the inhibition of mitochondrial fission by NS4B, which prevents the onset of mitochondrial antiviral status due to decreased ROS production [25,26]. Additionally, illustrating the deleterious effect of aerobic glycolysis on infected cells, the lactate produced by this process has been shown to affect RIG-I-dependent IFN production by regulating MAVS [27].

It has been stated on several occasions that the supply of TCA, and therefore of OXPHOS, can occur through the anaplerotic use of glutamine to give α-ketoglutarate. However, to date, the beneficial effect of glutamine during infection remains unclear. Indeed, it has been suggested that infected cells may be ‘addicted’ to glutamine, although it has not been established whether this addiction is due to the virus or rather to the requirement for glutamine to mount an effective antiviral response [28]. The work of Shiratori et al. illustrates that the use of galactose to force the metabolic reprogramming of cancer cells toward OXPHOS relies on increasing the glutamine pool to anaplerotically generate TCA intermediates [29]. In light of these data and our previous work, we hypothesize here that the host cell depends on glutamine to establish a metabolic process adapted to the development of an effective antiviral response. To test this hypothesis, we decided to evaluate the antiviral response in the presence or absence of L-glutamine in the culture medium. Interestingly, we demonstrated the existence of a bivalent antiviral response depending on cellular metabolism. Indeed, some ISGs are more expressed in an OXPHOS-dependent manner, as we expected, while others are more expressed in a glycolytic-dependent manner, the dichotomy in part relying on a variation of IFN-β expression.

## 2. Materials and Methods

### 2.1. Cells, Viruses and Reagents

The human A549-Dual™ (A549^Dual^) cells are adherent epithelial cells derived from A549 lung carcinoma cell line (ATCC, CCL-185™). A549^Dual^ cells were purchased from InvivoGen (ref. a549d-nfis, San Diego, CA, USA). Zika virus (PF-25013-18) and dengue virus serotype 2 (UVE/DENV-2/UNK/RE/CNR 14445) were amplified on Vero cells (ATCC, CCL-81) and viral titers were determined by a standard plaque-forming assay as previously described with minor modifications [30]. QUANTI-Luc™ substrate was obtained from InvivoGen (ref. rep-qlc1, San Diego, CA, USA). Oligomycin (ref. 75351), myxothiazol (ref. T5580), 2-deoxyglucose (2-DG, ref. D8375), 2,4-dinitrophenol (DNP, ref. D198501) and rotenone (ref. R8875) were purchased from Sigma-Aldrich (Saint-Louis, MO, USA). Poly(I:C) (HMW)/LyoVec™ (PIC) was purchased from InvivoGen (ref. tlrl-piclv, San Diego, CA, USA). MitoSOX™ Red was purchased from InvitroGen (ref. M36008, Carlsbad, CA, USA). Non-essential amino acid solution was purchased from PAN Biotech (ref. P08-32100, Aidenbach, Germany).

### 2.2. Cell Culture

The A549^Dual^ cells were cultured under standard maintenance conditions at 37 °C and 5% CO_2_ in Dulbecco’s Modified Eagle’s Medium (DMEM; PAN Biotech, Aidenbach, Germany) containing 4.5 g/L glucose, and supplemented with 10% heat-inactivated fetal bovine serum (FBS, Dutscher, Brumath, France), 100 μg·mL^−1^ streptomycin, 2 mM L-glutamine, 100 U·mL^−1^ penicillin and 0.5 µg/mL amphotericin B (PAN Biotech, Dutsher, Brumath, France). The culture medium also included 100 µg·mL^−1^ zeocin and 10 µg·mL^−1^ blasticidin (InvivoGen, San Diego, CA, USA) for the selection of cells expressing NF-κB-SEAP and IRF-Luc reporter proteins.

### 2.3. Oxygen Consumption Rate Assessment

A549^Dual^ cells were cultured in 25 cm^2^ flasks and DMEM 4.5 g·L^−1^ glucose was replaced by RPMI 2 g·L^−1^ glucose, supplemented or not with 2 mM L-glutamine, either containing PIC or not, 24 h before oxygen consumption rate (OCR) assessment. The cells were then washed with PBS, collected with trypsin/EDTA and centrifuged at 800× *g* for 5 min at room temperature. Subsequently, cell pellets were suspended in a volume of RPMI 2 g·L^−1^ glucose, supplemented or not with 2 mM L-glutamine, allowing the number of cells per chamber to be standardized (approximately 2 × 10^6^ cells). Basal oxygen consumption rates were continuously measured on whole cells at 37 °C, using a Clark-type oxygen electrode (Oroboros Oxygraph-2k, Oroboros Instruments, Innsbruck, Austria). For calibration of the polarographic oxygen sensor and measurement of background oxygen consumption, the medium without biological sample was added to the O2k-chamber. For each experimental condition, OCR was measured in the presence or absence of the F1F0-ATP synthase inhibitor oligomycin (1 μM), the glycolysis inhibitor 2-DG (5 mM), the protonophoric uncoupler dinitrophenol (100 μM) and a mix of the mitochondrial complex III inhibitor myxothiazol (1 µM) and the mitochondrial complex I inhibitor rotenone (1 μM). Data were expressed as picomoles O_2_ consumed by cells per second and per million cells.

### 2.4. Cytotoxicity Assay

A549^Dual^ cells were seeded in a 96-well plate at a density of 10,000 cells per well. The cells were treated with RPMI containing 2 g·L^−1^ glucose, with or without 2-mM L-glutamine and increasing doses of 2-DG (ranging from 1 mM to 20 mM). DMEM—DMSO 10% was used as a positive death inductor.

After 24 h, cell viability was assessed using the neutral red uptake assay, as described by Repetto et al. [31]. Briefly, after removing the culture supernatant, 100 µL of medium containing 40 µg·mL^−1^ of neutral red was added to each well, and the cells were incubated for 2 h at 37 °C. The medium was then removed, and the cells were washed with PBS. Finally, 150 µL of neutral red destain solution (1% glacial acetic acid, 49% water, 50% ethanol) was added to each well. Absorbance was measured at 540 nm using FLUOstar^®^ Omega microplate reader (BMG LABTECH, Offenburg, Germany).
$$Cell\;viability\;(\%)=100\times \frac{{Experimental\;condition\;(OD}_{540})}{Negative\;control\;{(OD}_{540})}$$

### 2.5. Cell Stimulation or Infection

The antiviral response was triggered by using polyriboinosinic:polyribocytidylic acid (PIC), a double-stranded RNA molecule that was combined with transfection agents to enhance its cellular uptake. A549^Dual^ cells were grown in RPMI-1640 medium (ref. P04-17550, PAN Biotech, Aidenbach, Germany) containing 10% FBS, supplemented with glucose and varying concentrations of L-glutamine (0 mM, 0.2 mM, 1 mM or 2 mM), in order to induce metabolic shift. After 24 h for adaptation, the cells were stimulated with 0.25 µg·mL^−1^ of PIC for 6 h to investigate the early stages of the IFN pathway, and for 24 h to study the late stages, including the production of ISGs. Similarly, the impact of inhibiting cellular respiration or glycolysis on the antiviral response was evaluated. A549^Dual^ cells were treated for 24 h with 0.25 µg·mL^−1^ PIC, along with either 1 µg·mL^−1^ rotenone to inhibit respiration or 5 mM 2-DG to inhibit glycolysis. Cells were cultured in RPMI-1640 containing 2 g·L^−1^ glucose, with or without 2 mM L-glutamine, for 24 h prior to the stimulation. After stimulation, the cell culture supernatant was collected and stored at −20 °C, while cells were either used fresh in flow cytometry, fixed for immunofluorescence or lysed using RLT Lysis Buffer (Qiagen, Hilden, Germany) for subsequent RNA extraction or using RIPA buffer for subsequent protein preparation.

Cell infection was achieved on A549^Dual^ cells with Zika virus (ZIKV) and dengue virus (DENV). Briefly, 24 h after seeding, medium was changed to RPMI-1640 either containing or not containing L-glutamine. After 24 h of adaptation, cells were infected for 24 h in RPMI-1640 supplemented or not with L-glutamine with ZIKV and DENV at a multiplicity of infection (MOI) of 2. Cells infected in conventional medium (e.g., complete DMEM) were used as control of infection. Following infection, total protein extract was harvested using RIPA buffer. Infection was monitored by detection of viral envelope (E) protein using a panflaviviral anti-E antibody (4G2) in western blot. Cell culture supernatant was harvested for luciferase activity assessment.

### 2.6. RT-qPCR

Total RNA has been extracted from cell lysates by using the RNeasy Plus Mini Kit (cat. 74136, Qiagen, Hilden, Germany). The total cDNA was obtained by reverse transcription using random primers from Invitrogen (ref. 58875, ThermoFisher, Waltham, MA, USA) and M-MLV reverse transcriptase enzyme (ref. M1708, Promega, Madison, WI, USA) at 42 °C for 60 min. cDNA was then subjected to a quantitative polymerase chain reaction, using a CFX96 Connect™ Real Time Detection System (Bio-Rad, Hercule, CA, USA). For amplification, BlasTaq™ 2X qPCR MasterMix (ref. G892-1, Applied Biological Materials, Vancouver, BC, Canada) and specific primers were used to assess gene transcripts expression for following targets (Table 1): ISG56, ISG54, ISG15, ISG12, IRF1, viperin, OAS1, CMPK2, lncRNA-NRIR, IFN-β, RNA polymerase II and 18S ribosomal RNA. A threshold cycle (Ct) was calculated for each single sample amplification reaction in the exponential phase of amplification, using Bio-Rad CFX Manager 3.1 (Bio-Rad, Hercule, CA, USA). Data obtained were expressed as mean normalized expression, with RNA polymerase II and 18S ribosomal RNA serving as housekeeping genes.

### 2.7. Western Blot

Stimulated cells (see Section 2.4. Cell stimulation) were lysed in 200 µL of radioimmunoprecipitation assay buffer (RIPA buffer) at 4 °C for 10 min. Lysates were sonicated at 0 °C for 1 min of pulse at 40% at 0.5 cycle using an Omni-Ruptor 4000 (Omni International, Dutscher, Issy-les-Moulineaux, France). Proteins from total cell extract were quantified using BCA reagent according to manufacturer instructions (Sigma-Aldrich, Humeau, La Chapelle-Sur-Erdre, France). A standardized amount of protein extracts was treated with Laemmli buffer containing 10 mM DTT and heated at 95 °C for 5 min. A total of 15 μg of proteins were separated on precast NuPAGE gels 4–12% (Thermofischer, Villebon-sur-Yvette, France) at 200 V constant for 35 min. Transfer was performed on NC Protan supported membrane with 0.45 μm pores (Amersham, GE, Buc, France) at 60 mA constant per gel for 2 h using a semi-dry system. After transfer, the membranes were saturated with 5% nonfat dry milk in PBS—Tween 0.1%. Membranes were then incubated successively with primary antibodies and secondary antibodies diluted in PBS—Tween 0.1%—1% nonfat dry milk (Table 2). Blots were revealed with enhanced chemiluminescence, ECL prime detection reagents (GE, Buc, France) using an Amersham Imager 680 (GE, Buc, France).

### 2.8. Mitochondrial ROS Measurement

Mitochondrial ROS were measured with MitoSOX™ Red (Invitrogen) using a flow cytometry-based protocol [32]. Briefly, A549^Dual^ cells were washed with PBS 1X, harvested with trypsin/EDTA and centrifuged at 800× *g* for 5 min. After three washes, cells were resuspended in preheated RPMI 2 g·L^−1^ glucose without phenol red, supplemented or not with 2 mM L-glutamine. MitoSOX™ Red at a final concentration of 5 µM was added to cell suspension and incubated for 30 min at 37 °C. After three washes with preheated medium, the expression of MitoSOX™ is quantified in the PE channel by flow cytometry using a CytoFLEX (Beckman Coulter, CA, USA).

### 2.9. Immunofluorescence

A549^Dual^ cells grown on glass coverslips were fixed with 3.7% paraformaldehyde—50 mM HEPES—150 mM NaCl at room temperature for 10 min. Fixed cells were permeabilized with 0.1% Triton X-100 in PBS for 3 min. The coverslips were incubated with primary antibodies in 1% BSA—PBS. Antigen—antibody binding was visualized with donkey anti-mouse Alexa 488 Fluor-conjugated and donkey anti-rabbit Alexa 488 Fluor-conjugated secondary antibodies (Table 3). Nucleus morphology was revealed by 4′,6-Diamidino-2-phenylindole dihydrochloride (DAPI) staining. The coverslips were mounted with ibidi Mounting Medium (ref. 50001, ibidi, Gräfelfing, Germany) and fluorescence was observed using an AXIO OBSERVER 7 equipped with the Apotome.2 (Zeiss, Oberkochen, Germany). The images were acquired with ZEN software version 3.1 (Zeiss, Oberkochen, Germany).

### 2.10. Measurement of the IFN-β Pathway Activation

The activation of the IFN pathway dependent on the Interferon regulatory factors (IRF) was assessed by measuring Lucia luciferase activity in A549^Dual^ cells. This was done using the QUANTI-Luc substrate following the manufacturer’s instructions. In brief, 100 µL of QUANTI-Luc substrate was added to 20 µL of cell culture supernatant, and the IRF-induced luciferase levels were quantified through luminescence measurement using a Luminoskan™ microplate luminometer (ThermoFisher, Waltham, MA, USA).

### 2.11. Transcription Factors Prediction

Prediction of potential binding sites for transcription factors in the promoter region of antiviral effectors was achieved using PROMO tool from ALGGEN group, based on TRANSFAC 8.3 [33,34]. Briefly, a sequence of at least 1000 bp upstream of the 1st exon of each of the following genes was recovered from the Ensembl database [35]: viperin (ENSG00000134321), OAS1 (ENSG00000089127), ISG12 (ENSG00000165949), ISG15 (ENSG00000187608) and ISG54 (ENSG00000119922). These sequences were submitted to MultiSearchSites to search for binding sites that viperin and OAS1 sequences share, on the one hand, and ISG12, ISG15 and ISG54 on the other, restricting the prediction to the human set of transcription factors. A Venn diagram was then drawn to illustrate the transcription factors common or specific to these groups [36].

### 2.12. Statistical Analyses

For all figures except Figure 1 and Figure 2C, statistical analyses were performed by two-way ANOVA with Dunnett’s correction using Graph-Pad Prism software version 9. For Figure 1 and Figure 2C, statistical analyses were done using unpaired *t*-test. Values of *p* < 0.05 were considered statistically significant. The degrees of significance are indicated in the figures captions as follows: * *p* < 0.0332; ** *p* < 0.0021; *** *p* < 0.0002; **** *p* < 0.0001.

## 3. Results

### 3.1. Enhancement of OXPHOS Occurs in Presence of L-Glutamine

We have previously demonstrated the ability of cells to produce a more effective antiviral response if cellular respiration was increased [22]. We then induced the metabolic reprogramming of A549^Dual^ cells by cultivating them in galactose. Galactose was shown to anaplerotically feed TCA via L-glutamine production [29]. It is established in the literature that L-glutamine is both essential for antiviral response and viral progeny [28]. An open question in the literature is whether this amino acid is more beneficial to the infected host cell or to the virus in this context. Also, it remains to be established which mechanisms underlie these observations. In light of our previous results, we hypothesized that L-glutamine would be essential for the establishment of an effective antiviral response, via the supply of TCA and OXPHOS.

As a first step, we therefore evaluated the ability of L-glutamine to induce an increase in cellular respiration. To do so, we cultured A549^Dual^ cells in RPMI containing 2 g·L^−1^ glucose, supplemented or not with 2 mM L-glutamine. After 24 h of adaptation, we measured the O_2_ consumption of the A549^Dual^ cells by oximetry, using different uncoupling agents. As shown in Figure 1A, the two culture conditions lead to a different O_2_ consumption rate. Indeed, the basal level of O_2_ consumption is significantly increased in the presence of L-glutamine (Figure 1B). Moreover, using 2,4-dinitrophenol to force the passage of protons from the mitochondrial intermembrane space reveals that the maximal respiratory capacity of A549^Dual^ cells is also significantly increased—40% higher—in the presence of L-glutamine (Figure 1C). Finally, we measured the proportion of O_2_ consumption not attributable to mitochondria, completely uncoupling the mitochondrial respiratory chain using a mixture of rotenone and myxothiazol. Cells grown without L-glutamine showed greater non-mitochondrial cellular respiration than cells grown in the presence of L-glutamine (Figure 1D). Notably, the proton leakage assessed through the use of oligomycin was not different between the two culture conditions. Also, the use of 2-deoxyglucose (2-DG) induces a decrease in O_2_ consumption but with no significant difference between cells cultured with or without L-glutamine. Thus, we have shown that L-glutamine induces an increase in cellular respiration in A549^Dual^ cells which, according to our hypothesis, could be associated with a greater antiviral response in an infectious context.
Figure 1OXPHOS is associated with L-glutamine-containing medium. (**A**) Comparison of the oxygen consumption rate of A549^Dual^ cells cultured with or without L-glutamine. A total of 5 mM 2-deoxyglucose (2-DG) has been used to inhibit hexokinase activity and glycolysis. A total of 1 µM oligomycin (Oligo) has been used to inhibit ATP synthase. A total of 100 µM 2,4-dinitrophenol (DNP) has been used as protonophoric uncoupler, allowing maximal respiration. A total of 1 µM Rotenone—1 µM Myxothiazol has been used to uncouple both mitochondrial complex I and III, illustrating non-mitochondrial respiration. The metabolism of cells grown in RPMI supplemented with 2 mM L-glutamine is more dependent on oxidative phosphorylation. More specifically, cells cultured with L-glutamine have higher basal O_2_ consumption flux (**B**), associated with higher maximal respiration capacity (**C**) and consistently lower non-mitochondrial respiration (**D**). Statistical analyses were done using unpaired *t*-test. Error bars represent standard deviation of three independent experiments. * *p* < 0.0332; **** *p* < 0.000. w/o L-Glut: without L-glutamine; w/L-Glut.: with 2 mM L-glutamine.
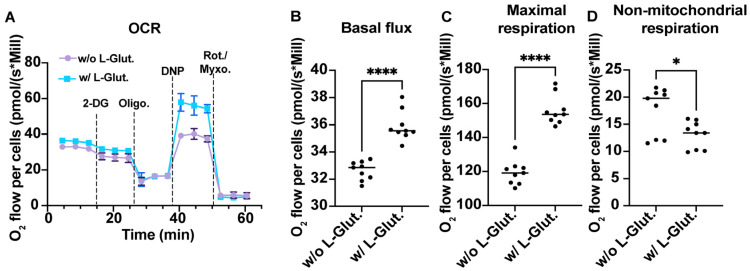


### 3.2. Antiviral Response Is Bivalent and Varies According to Metabolic Context

As mentioned above, the role of L-glutamine in infection is not well defined yet. Some believe that L-glutamine is essential for virus progeny, making the virus addicted to L-glutamine [28]. Others suggest that L-glutamine is necessary for the antiviral response, making the host cell addicted to L-glutamine during infection [37]. Assuming that L-glutamine supports a metabolism based on OXPHOS, it seems reasonable to suppose that it would also support the development of the antiviral response. In order to prove the host cell’s addiction to L-glutamine for the development of the antiviral response, we decided to use poly:IC (PIC) with increasing doses of L-glutamine. PIC is a synthetic double-stranded RNA commonly used as a pathogen-associated molecular pattern mimic. It is known to imitate replicating viral RNA and stimulate the antiviral response by activating TLR3, RIG-I and RIG-I-like receptors (RLR), leading to the production of type I IFNs and the expression of ISGs [38,39].

Thus, we cultured A549^Dual^ cells with increasing doses of L-glutamine (from 0 to 2 mM) for 24 h before stimulation with PIC 0.25 µg·mL^−1^. After 24 h of stimulation, we assessed the development of the antiviral response by measuring ISGs expression by RT-qPCR. All ISGs tested here (IRF1, ISG15, ISG56, ISG54 and ISG12) showed an increase in expression in response to PIC. Also, interestingly, their expression increases in response to L-glutamine, consistently with the level of expression found in cells cultured in complete DMEM. The difference in expression with cells cultured without L-glutamine was significant as early as 0.2 mM L-glutamine, with levels varying between 2- and 5-fold more, depending on the gene studied (Figure 2A). To validate the observed transcriptional effects, we evaluated the synthesis of proteins regulated by the ISRE promoter. Specifically, as we used A549^Dual^ cells, which express Lucia luciferase under the control of an ISG54 minimal promoter and five IFN-stimulated response elements (ISRE), we were able to assess the protein expression of ISGs by measuring the luciferase secreted into the cell culture supernatant. Accordingly, we examined the release of luciferase into the supernatant of cells treated with PIC by quantifying the emitted luminescence using the Lucia luciferase substrate (QUANTI-Luc). Consistent with the results obtained from RT-qPCR, we detected a significant increase in luciferase activity upon PIC stimulation and dose-dependently along with an increase in L-glutamine concentration (up to 10-fold), indicating a significant upregulation of antiviral proteins regulated by the ISRE promoter (Figure 2B).
Figure 2L-glutamine increases antiviral response to poly:IC. (**A**) L-glutamine-related overexpression of ISGs expression. The expression of several ISGs (IRF1, ISG15, ISG56, ISG54, ISG12) has been assessed by RT-qPCR after 24 h stimulation of A549^Dual^ cells, grown in RPMI 2 g·L^−1^ glucose, supplemented with increasing doses of L-glutamine (0, 0.2, 1 and 2 mM), using poly:IC (PIC) 0.25 µg·mL^−1^. Results are expressed as mean normalized expression, following normalization with expression of 18S ribosomal RNA as a housekeeping gene. (**B**) ISRE-dependent proteins increased production in presence of L-glutamine. Evaluation of IRF pathway activation in response to PIC 0.25 µg·mL^−1^ stimulation of A549^Dual^ cells grown in RPMI 2 g·L^−1^ glucose supplemented with increasing doses of L-glutamine (0, 0.2, 1 and 2 mM) was achieved by measuring activity of secreted Lucia luciferase, using QUANTI-Luc substrate 24 h after treatment. Results are expressed as emitted luminescence intensity. (**C**) mROS induction is higher in presence of L-glutamine. Mitochondrial ROS were measured with MitoSOX™ Red using a flow cytometry-based protocol in cells grown in RPMI 2 g·L^−1^ glucose, supplemented or not with 2 mM L-glutamine, after 6 h of stimulation with PIC. Statistical analyses were conducted using ANOVA with Dunnett’s correction (**A**,**B**) and unpaired *t*-test (**C**). Error bars represent standard deviation of at least three independent experiments. * *p* < 0.0332; ** *p* < 0.0021; *** *p* < 0.0002; **** *p* < 0.0001. w/o L-Glut: without L-glutamine; w/L-Glut.: with 2 mM L-glutamine.
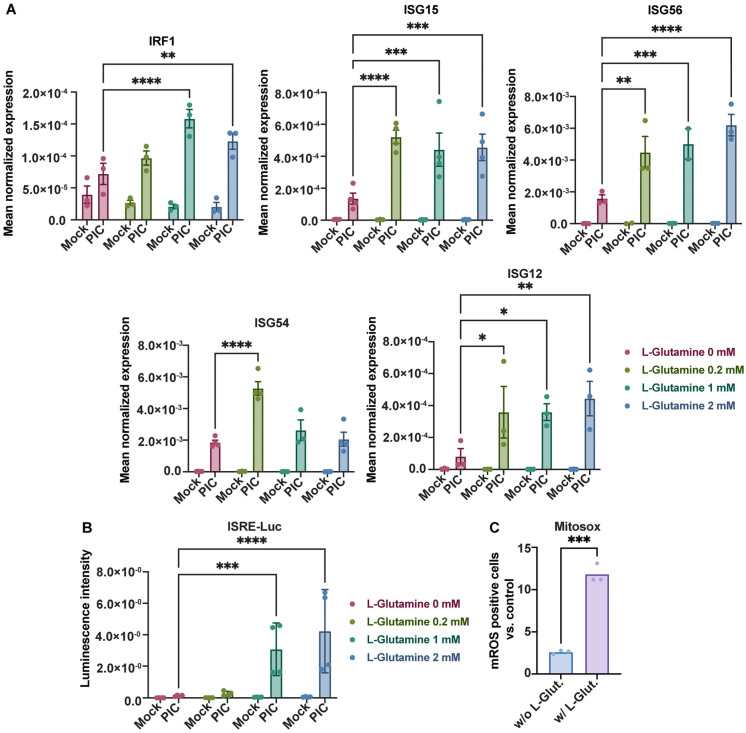


OXPHOS within the mitochondria has been regularly associated with the production of mitochondrial reactive oxygen species (mROS) that have several physiological functions [40]. Among the functions repeatedly associated with these mROS is the promotion of the antiviral response [23,41]. Using MitoSOX™, we explored the formation of mROS in our model by flow cytometry. Thus, after 6 h of stimulation with PIC, the level of cells becoming positive for mROS was greater in cells cultured with 2 mM L-glutamine when compared to cells cultured without L-glutamine, illustrating the transient production of mROS with the establishment of the early steps of the antiviral response (Figure 2C). Moreover, to confirm that antiviral response establishment is associated with increased mitochondrial activity and OXPHOS, we repeated high-resolution respirometry measurements after PIC stimulation (Appendix A). As illustrated, PIC stimulation is linked to a higher OXPHOS in A549^Dual^ cells cultured in 2 mM L-glutamine versus cells cultured without L-glutamine, whether basally or at maximum respiratory capacity, reinforcing the link between this metabolic state and higher expression of antiviral effectors. These results were associated with increased TOM20 labeling in A549^Dual^ cells cultured in L-glutamine and treated with PIC for 6 h (Appendix A). The increase in the TOM20 signal is another manifestation of a more functional mitochondrial network in the presence of L-glutamine, supporting OXPHOS (Figure 1 and Appendix A) and the induction of mROS (Figure 2C).

Since a stimulation with PIC, as we have conducted, cannot summarize all aspects of infection, we endeavored to deepen our work demonstrating that such a dependency on L-glutamine exists even during a viral infection. Thus, we employed two RNA viruses from the *Flavivirus* family (the Zika virus and dengue virus) and were able to demonstrate that during infection with these viruses (at MOI 2), the antiviral response is significantly affected by the presence of glutamine in the cell culture medium.

As illustrated in Appendix A, the antiviral response assessed here thanks to the A549^Dual^ reporter system shows a dose-dependent increase linked to the addition of glutamine, thus illustrating the dependency of this cellular response on OXPHOS, even during viral infection.

Furthermore, to confirm that the increase in antiviral response is related to a specific effect of L-glutamine, cells were stimulated with PIC 0.25 µg·mL^−1^ in the presence (or absence) of a solution of non-essential amino acids, NEAA (0.1 mM L-alanine, 0.1 mM L-asparagine, 0.1 mM L-aspartic acid, 0.1 mM L-glutamic acid, 0.1 mM glycine, 0.1 mM L-proline and 0.1 mM L-serine), to which 2 mM L-glutamine was added (or not). As shown in Appendix A, luciferase activity, and therefore the antiviral response it reflects, was not increased in the presence of NEAA alone. Notably, the addition of L-glutamine to this cocktail restored the previous results.

Surprisingly, the expression of some ISGs does not follow this trend; it is quite the contrary. Therefore, by evaluating the expression of viperin and 2′-5′-oligoadenylate synthetase 1 (OAS1) by RT-qPCR under the same conditions, we established that these genes are more highly expressed at a low L-glutamine concentration. Indeed, their expression decreases inversely (up to 7-fold) with an increasing L-glutamine concentration (Figure 3). This suggests that there is in fact a dichotomy in the expression profile of ISGs depending on the predominant metabolic pathway at the time of PIC stimulation. If OXPHOS is favored, through L-glutamine, there will be a greater expression of ISG12, ISG15, ISG54, ISG56 and IRF1. On the contrary, if glycolysis is promoted, in this case in the absence of L-glutamine, other ISGs will be favored, such as OAS1 and viperin. To test the accuracy of this hypothesis, we needed to inhibit these different metabolic pathways under infectious conditions and evaluate the development of the antiviral response for the different mentioned ISGs.

### 3.3. Antiviral Response Is Bivalent and Relying Alternatively on Glycolysis or OXPHOS

As before [22], we used rotenone at a concentration of 1 µg·mL^−1^ to inhibit OXPHOS, since rotenone is a broad-spectrum piscicide, pesticide and insecticide derived from plants, which is known to inhibit complex I of the mitochondrial electron transport chain, thereby blocking oxidative phosphorylation and the associated production of ATP [42], while 2-DG at 5 mM was used to inhibit glycolysis. Indeed, 2-DG is a glucose molecule with a 2-hydroxyl group replaced by hydrogen, leading to a molecule unsuitable for glycolysis which competitively inhibits glucose-6-phosphate production by phosphoglucoisomerase [43]. Notably, at the dose of 5 mM, 2-DG exerts no cytotoxic effect, assessed by a neutral red uptake assay, on A549^Dual^ cells as shown in Appendix A. Thus, we used these inhibitors in combination with stimulation by 0.25 µg·mL^−1^ PIC to study the impact of glycolysis and OXPHOS inhibition on the expression of the antiviral genes presented above.

First, we assessed this expression at the gene level using RT-qPCR. As before, the expression of IRF1, ISG12, ISG15, ISG54 and ISG56 was significantly increased following stimulation with PIC in the presence of 2 mM L-glutamine compared with cells cultured without L-glutamine (Figure 4). On the contrary, viperin and OAS1 were expressed more when cells were cultured without L-glutamine. The addition of 2-DG, and the subsequent inhibition of glycolysis, resulted in no significant difference in the expression of IRF1, ISG12, ISG15 and ISG56. On the other hand, the expression of OAS1 and viperin is significantly altered in cells cultured without L-glutamine when glycolysis is inhibited (Figure 4). Therefore, we can conclude that, at the transcriptional level, the expression of genes such as OAS1 and viperin appears to be dependent on glycolysis, whereas the expression of other genes (e.g., ISG12, ISG15, IRF1) is independent of this metabolic pathway. Interestingly, the expression of these same genes is significantly inhibited in the presence of rotenone when cells are cultured with L-glutamine. Their expression level even reached that of cells cultured without L-glutamine, especially for ISG54 (Figure 4). Regarding viperin, its expression appeared less sensitive to rotenone in cells cultured without L-glutamine, illustrating that this gene is less dependent on OXPHOS than the others (Figure 4). Thus, we have demonstrated here that, at the transcriptional level, the increased expression of certain antiviral genes, such as ISG54 and ISG15, in L-glutamine-cultured cells is related to the higher OXPHOS rate in these cells.

At the protein level, we again studied the luminescence emitted by secreted luciferase in response to PIC to estimate the production of antiviral proteins under the control of the ISRE promoter. The results obtained are consistent with those obtained at the gene level for IRF1, ISG12, ISG15, ISG54 and ISG56. Namely, a decrease in luciferase activity, reflecting a reduction in the production of viral proteins under ISRE control, in response to rotenone treatment (Figure 5A). Finally, by Western blot, we found a higher signal for ISG56 and ISG15 in response to PIC for cells cultured in L-glutamine compared to cells cultured without (Figure 5B). This signal decreased after rotenone treatment, indicating that the upregulated production of these antiviral proteins in L-glutamine-grown cells is indeed related to higher OXPHOS. Viperin, also assessed by Western blot, follows an opposite pattern. Indeed, cells cultured without L-glutamine show a higher signal in response to PIC, illustrating greater viperin production in these cells than in cells cultured with L-glutamine. However, this production was reduced after the inhibition of glycolysis by 2-DG (Figure 5B).

### 3.4. Viperin Is Expressed in a Type I IFN-Independent Manner in Glycolytic Condition

In an attempt to explain the specificity of viperin expression, we hypothesized that the viperin promoter region could possibly contain an element enabling greater viperin expression in response to glycolysis. Therefore, we evaluated the expression by RT-qPCR of cytidine/uridine monophosphate kinase 2 (CMPK2) and the lncRNA-negative regulator of the IFN response (lncRNA-NRIR), both of which are expressed at the same time as viperin (Figure 6A) [44]. As shown in Figure 6B, the expression of CMPK2 or lncRNA-NRIR did not vary between L-glutamine and L-glutamine-free conditions. This result corroborates the fact that OAS1 also responds better in glycolysis, even though it does not depend on the same promoter region as viperin.

We then attempted to determine whether the switch that leads to a dichotomy in the ISGs expression profile between cells cultured in L-glutamine versus non-L-glutamine cells could be related to a difference in the expression of type I IFNs between the two conditions. Thus, by RT-qPCR, we assessed IFN-β expression in response to a 6 h stimulation with PIC, in the presence or absence of L-glutamine. Interestingly, cells cultured without L-glutamine exert an expression of IFN-β similar to that observed for cells cultured with 2 mM L-glutamine (Figure 6C). However, this expression is not converted to the protein level, since the signal found for IFN-β is weaker for cells cultured without L-glutamine, whether assessed by Western Blot (Figure 6D) or immunofluorescence (Figure 6E). The higher expression of IFN-β in cells cultured with 2 mM L-glutamine results in greater STAT2-mediated signaling in these cells compared to cells cultured without L-glutamine, as evidenced by the stronger signal observed for pSTAT2 for this condition (Figure 6D). Therefore, a higher expression of ISGs such as ISG54, ISG15 or ISG56 would be associated with more efficient IFN-dependent signaling in the presence of L-glutamine. On the other hand, the increase in viperin and OAS1 expression in glycolysis appears to be independent of IFN signaling. At this point, we still assume that it is possible that the higher expression of these two genes in response to PIC under glycolytic conditions may result from the activation of a transcription factor specific to these two genes compared to the other genes studied here. We therefore sought possible binding sites for transcription factors found in common between the viperin and OAS1 genes on the one hand, and ISG15, ISG54 and ISG12 on the other. To do this, we analyzed a sequence of more than 1000 bp upstream of the first exon of each of these genes, using the PROMO tool, based on TRANSFAC 8.3. We then compared the reported binding sites between these two analysis groups (Figure 6F). As shown in the Venn diagram, fifty-eight potential binding sites are found to be in common between viperin and OAS1, but only three of these sites are found to be specific to these genes compared to the other ISGs tested: STAT5A, NF-Y and E2F-1. Therefore, it is possible that one of these three transcription factors is responsible for the expression of viperin and OAS1 in cells treated without L-glutamine.

## 4. Discussion

With the succession of viral epidemics, some of which have a global impact, research efforts to understand host response mechanisms have intensified. While much progress has been made in understanding the role of metabolism in the activation and function of immune cells, and therefore in understanding how these cells respond to viral infection depending on their metabolic setting, very little is known about the innate antiviral response of the infected host cell. An interesting question is whether their metabolism influences the canonical antiviral response, namely the IFN response. 

It has been established on several occasions that, during viral infection, viruses manipulate host cell metabolism, diverting them from OXPHOS to glycolysis [21]. This detour has been shown to be of particular interest in viral progeny, as in the case of HIV and DENV [24,45,46]. However, the influence of such hijacking on the production of antiviral effector molecules induced by the IFN response has not yet been explored. The role of glutaminergic metabolism in the antiviral response also remains unresolved in the literature; does the benefit fall to the virus or to the host cell in the event of infection [28]? We have previously shown that OXPHOS promotes the expression of certain antiviral effectors such as ISG54 and ISG56. Furthermore, given that glutaminergic metabolism is closely linked to OXPHOS through the anaplerotic feeding of TCA, we hypothesized here that glutamine intake in an infectious context would increase the antiviral response and the production of antiviral effectors. To address this hypothesis, we stimulated A549^Dual^ cells with PIC in the presence or absence of L-glutamine in order to study the intrinsic effect of metabolism on the antiviral response, while avoiding the inhibitory effect that viral proteins can induce during infection. 

Here, we were able to demonstrate that cells cultured with 2 mM L-glutamine had more functional mitochondria (Figure 1 and Appendix A), which supported a greater induction of OXPHOS and mROS (Figure 1, Figure 2C and Appendix A). Furthermore, the increased mitochondrial activity in these cells was correlated with a greater production of certain antiviral effectors (ISG12, ISG15, ISG54, ISG56, IRF1), as shown by the inhibition of their expression in the presence of rotenone (Figure 2, Figure 4 and Figure 5). Moreover, this increased antiviral state in the presence of glutamine was observed during viral infection using ZIKV and DENV (Appendix A). However, we were also able to show that the expression of certain ISGs escapes the need for OXPHOS, since viperin and OAS1 are more highly expressed in the presence of aerobic glycolysis (Figure 3, Figure 4 and Figure 5). One factor that may explain this dichotomy is that IFN-β expression and subsequent signaling are greater in cells to which L-glutamine has been added (Figure 6), while the higher expression of viperin and OAS1 could be related to a specific transcription factor such as E2F-1 (Figure 6F).

It had previously been shown that viperin expression was increased under conditions of high glucose and therefore glycolysis, enabling the control of Zika virus infection [47]. These results were surprising in light of our work illustrating the beneficial effect of OXPHOS on the production of other antiviral effectors such as ISG54 and ISG56 [22]. Our study reconciles these results, as we have demonstrated the existence of a bivalent antiviral response, the orientation of which depends on cellular metabolic status (Figure 7). Therefore, OXPHOS in L-glutamine culture favors the production of ISGs such as ISG12, 15, 54 and 56, whereas glycolysis is beneficial for the production of viperin and OAS1. These results suggest that the bivalence of the antiviral response may be of interest in the event of metabolic reprogramming by the virus, with one taking over from the other. Thus, in the case of a Warburg-like effect, the host cell can still base its antiviral response on the induction of potent antiviral effectors such as viperin or OAS1. The results presented here stem from experiments carried out on A549 cells, which are cancer cells. While A549 cells can reproduce many of the physiological processes that occur during infection, we recognize that their metabolism differs from normal physiology, and we have sought to address this through our study model. Although these results require confirmation using primary cells, elements in the literature support the validity of these findings. Several studies have corroborated our findings using different cell types. For instance, as previously mentioned, Reslan et al. showed that a glycolytic state in proximal tubular cells (HK-2) favored viperin expression [47]. Additionally, supporting the idea that oxidative phosphorylation promotes the type I IFN response, Buskiewicz et al. demonstrated in mouse embryonic fibroblasts and human peripheral blood mononuclear cells that the inhibition of the formation of mitochondrial ROS limits MAVS oligomerization and therefore type I IFN secretion [23]. Finally, in a recent study conducted with macrophages, Waqas et al. found that ISG15 deficiency was associated with decreased glutamine metabolism and reduced OXPHOS [48]. These findings support the interplay between ISGs, glutamine metabolism and mitochondrial activity reported here.

Furthermore, our results showed that the promotion of OXPHOS by glutamine favored the expression of antiviral effectors, even in a context of total infection, using dengue and Zika viruses. The results we obtained are consistent with what is already known for these viruses. During infection, DENV and ZIKV showed an early capacity to divert cellular metabolism towards glycolysis. Thus, at 10 h.p.i., an increase in lactate produced by DENV-infected cells was reported [45,49]. In addition, a significant decrease in OXPHOS was demonstrated at 24 h.p.i. in infected HUH7 [50]. Similar results were obtained for ZIKV, with a 12 h.p.i. decrease in OXPHOS [51]. The diversion of cellular metabolism towards glycolysis clearly favors the virus during the early phases of its viral cycle, on the one hand by enabling the production of biomass through a Warburg-like effect, but also, as we have been able to demonstrate with our results, by limiting the development of the antiviral response. However, as demonstrated in the literature, this process of diversion cannot last throughout the infection. Indeed, later energy-intensive processes, such as the transition through the secretory pathway, may require the production of a large amount of energy, which explains the increase in OXPHOS observed at 48 h.p.i. [52]. Control of the antiviral response due to OXPHOS in this phase of infection would therefore be achieved by the regulation of endoplasmic reticulum–mitochondria interfaces as demonstrated by the ability of DENV NS4B to regulate mitochondrial fission by the inhibition of DRP1, leading to mitochondria-associated membrane disruption and thus the subsequent inhibition of the RIG-I-dependent antiviral response [25]. The regulation of cellular metabolism by these viruses during infection, therefore, seems to be associated with precise mechanisms, the sequence of which is always designed to promote viral replication and progeny. Promoting OXPHOS and the antiviral response that may be associated with it could therefore be beneficial in limiting viral replication. However, the question of glutamine dependence may arise and seems to affect ZIKV and DENV rather than the host, since these viruses do not seem to be able to replicate in the absence of glutamine (as shown in Appendix A). In this case, should we favor the OXPHOS-dependent antiviral response (ISG15, 56 and 54)? Or should we limit glutamine intake, which would inhibit viral replication while allowing the expression of glycolysis-dependent ISGs (viperin and OAS1)? In particular, it should be mentioned that the study of the influence of viral infection on metabolism is complex. The data currently available in the literature illustrate a variation in the effect of viral infection on metabolism depending on the infected host. This is particularly true for the Zika virus, which behaves differently as it infects its vector, the mosquito [53]. An important aspect that remains to be studied is the impact of this variability in antiviral response on viral replication. However, as both of these aspects rely on glutaminergic metabolism, it could prove complicated to delineate the correlation between different metabolic states, ISG expression and viral growth kinetics as well as peak titers in the context of glutamine availability. Should the eventual reduction in viral titer be attributed to the antiviral response or to altered glutaminergic metabolism? This essential question needs to be answered through future studies.

However, through this work, we have not been able to uncover the mechanism leading to the greater expression of viperin in a glycolytic context. And this important aspect remains to be explored in future work. We hypothesize that the expression of viperin and OAS1 could be linked to the activation of a specific transcription factor in this context. Among those we have identified (Figure 6F), it is known that E2F-1 is normally sequestered by the retinoblastoma protein (Rb). In the presence of glucose, the E2F-1/pRB axis is more active, leading to the release of E2F-1 and transcription of its targets. However, everything may not be solely linked to this transcription factor, and we cannot rule out that it may be a combination of various factors or even a specific timing that promotes the expression of viperin and OAS1 during aerobic glycolysis. Additionally, it is possible to argue that viperin is known to be expressed independently of interferon in a mode involving NF-κB. These hypotheses should be explored in future studies. Finally, we have been able to demonstrate that OXPHOS promotes IFN-β production, in contrast to glycolysis, leading to a more efficient production of antiviral effectors.

In light of our initial results, it might be possible to argue that the glutamine addiction discussed in the literature initially seems to go to the host cell, since it produces more antiviral effectors in the presence of L-glutamine. However, since some antiviral effectors can overcome this dependence, it seems that glutamine is more essential to the virus for its progeny [28]. This suggests an adaptive mechanism, whereby, to counteract the virus-induced metabolic reprogramming that undermines the antiviral response carried out by ISGs (12, 15, 54 and 56), the host cell has found a way to promote the production of other effectors such as viperin and OAS1. We could be in a frenetic adaptive race in which the cellular metabolic status is of prime importance. These results are encouraging and suggest that the study of glutaminergic metabolism could be an interesting way of approaching the control of a viral infection and thus exploring possible therapeutic or preventive opportunities. However, these data still need to be consolidated and further experiments on animal models should be considered to confirm these mechanisms in vivo. There is some evidence to suggest that OXPHOS in vivo is favorable, since the pattern presented here is reminiscent of the antiviral response in bats. Indeed, these mammals are the only ones to have acquired the ability to fly, which is associated with a high respiratory capacity and, consequently, high ROS production. This favors the interferon response in these mammals and confers on them the ability to control most of the pathogens that infect them, limiting the associated pathophysiology [54,55,56].

## Figures and Tables

**Figure 3 viruses-16-01391-f003:**
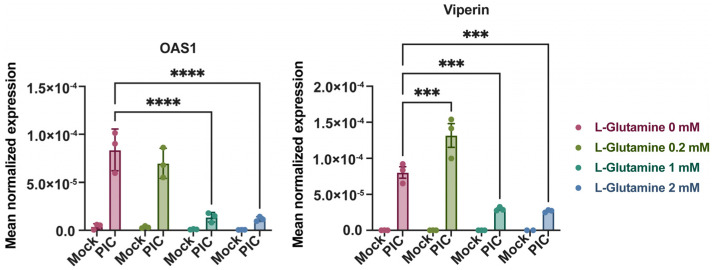
Expression of OAS1 and viperin are independent of L-glutamine. Expression of OAS1 and viperin was assessed by RT-qPCR following 24-h stimulation of A549^Dual^ cells, grown in RPMI 2 g·L^−1^ glucose, supplemented with increasing doses of L-glutamine (0, 0.2, 1 and 2 mM), using poly:IC (PIC) 0.25 µg·mL^−1^. Results are expressed as mean normalized expression, following normalization with expression of 18S ribosomal RNA as housekeeping gene. Statistical analyses were conducted using ANOVA with Dunnett’s correction. Error bars represent standard deviation of at least three independent experiments. *** *p* < 0.0002; **** *p* < 0.0001.

**Figure 4 viruses-16-01391-f004:**
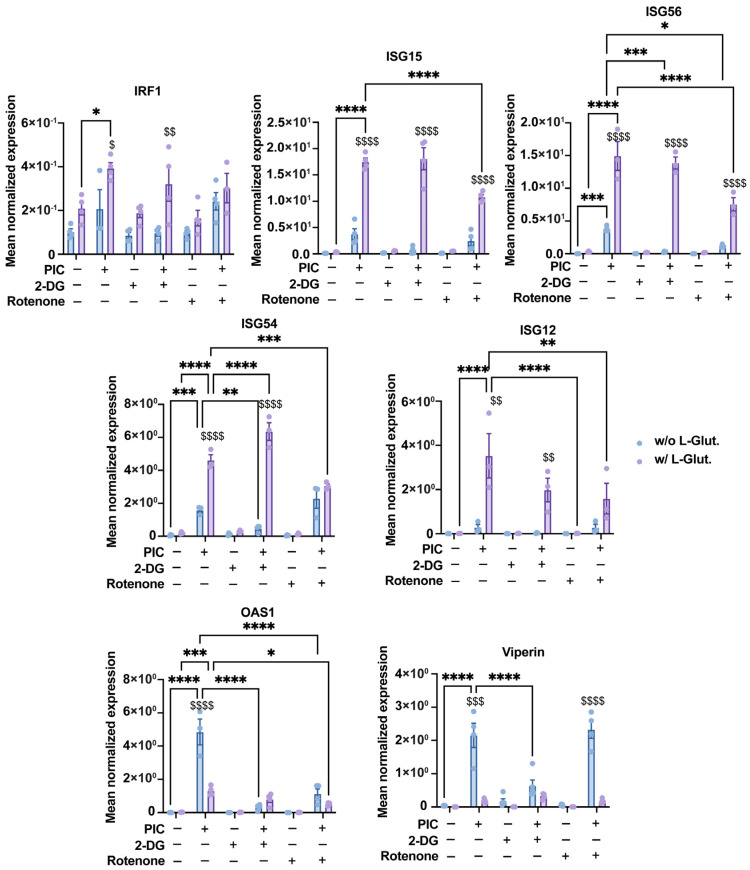
Bivalent expression of ISGs is based on glycolysis and OXPHOS. Expression of several ISGs (IRF1, ISG15, ISG56, ISG54, ISG12, OAS1 and viperin) has been assessed by RT-qPCR following a 24 h stimulation of A549^Dual^ cells, grown in RPMI 2 g·L^−1^ glucose, supplemented or not with 2 mM L-glutamine, using poly:IC (PIC) 0.25 µg·mL^−1^. Simultaneously, to evaluate influence of glycolysis and OXPHOS on upregulation of ISGs regarding L-glutamine supplementation, 5 mM 2-deoxyglucose (2-DG) and 1 µg·mL^−1^ rotenone were used to inhibit glycolysis and OXPHOS, respectively. Results are expressed as mean normalized expression, following normalization with expression of RNA polymerase II RNA as housekeeping gene. Statistical analyses were conducted using ANOVA with Dunnett’s correction. Error bars represent standard deviation of at least three independent experiments. * *p* < 0.0332, ** *p* < 0.0021, *** *p* < 0.0002, and **** *p* < 0.0001 as compared to PIC. ^$^ *p* < 0.0332, ^$$^ *p* < 0.0021, ^$$$^ *p* < 0.0002, and ^$$$$^ *p* < 0.0001 as compared to cells cultured without L-glutamine vs. those with L-glutamine. w/o L-Glut: without L-glutamine; w/L-Glut.: with 2 mM L-glutamine.

**Figure 5 viruses-16-01391-f005:**
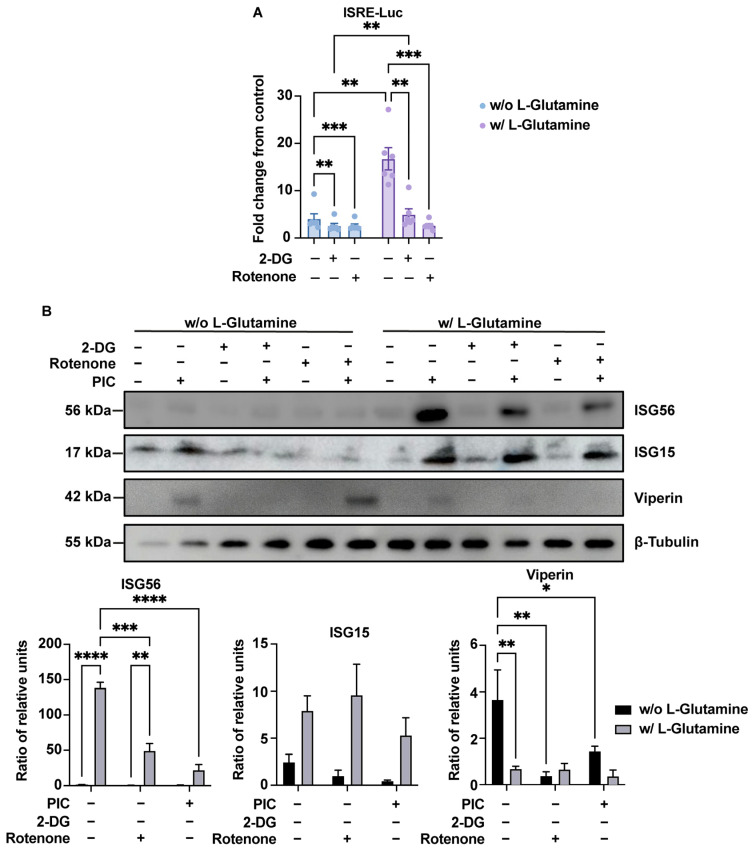
Bivalent ISG production depends on glycolysis or OXPHOS. (**A**) ISRE-dependent proteins increased production in presence of L-glutamine is linked to OXPHOS. Evaluation of IRF pathway activation in response to poly:IC (PIC) 0.25 µg·mL^−1^ stimulation of A549^Dual^ cells grown in RPMI 2 g·L^−1^ glucose, supplemented or not with 2 mM L-glutamine, was achieved by measuring activity of secreted Lucia luciferase, using QUANTI-Luc substrate 24 h after treatment. Concomitantly, to assess the influence of glycolysis and OXPHOS on the upregulation of ISGs with respect to L-glutamine supplementation, 5 mM 2-deoxyglucose (2-DG) and 1 µg·mL^−1^ rotenone were used to inhibit glycolysis and OXPHOS. Results are expressed as fold change in emitted luminescence intensity from mock stimulated cells. Statistical analyses were conducted using ANOVA with Dunnett’s correction. Error bars represent standard deviation of at least three independent experiments. ** *p* < 0.0021; *** *p* < 0.0002. (**B**) Bivalent production of ISGs relies on either glycolysis or OXPHOS. ISGs production in response to PIC 0.25 µg·mL^−1^ stimulation of A549^Dual^ cells grown in RPMI 2 g·L^−1^ glucose, supplemented or not with 2 mM L-glutamine, was assessed by western blot analysis. Western blot assay was achieved with mouse anti-ISG56, rabbit anti-ISG15, mouse anti-viperin and rabbit anti-β-tubulin antibodies. These images are representative of three independent experiments. Bands density was evaluated and signal emission normalized to protein loading. Statistical analyses were conducted using ANOVA with Dunnett’s correction. Error bars represent standard deviation of at least three independent experiments. * *p* < 0.0332; ** *p* < 0.0021; *** *p* < 0.0002; **** *p* < 0.0001. w/o L-Glutamine: without L-glutamine; w/L-Glutamine: with 2 mM L-glutamine.

**Figure 6 viruses-16-01391-f006:**
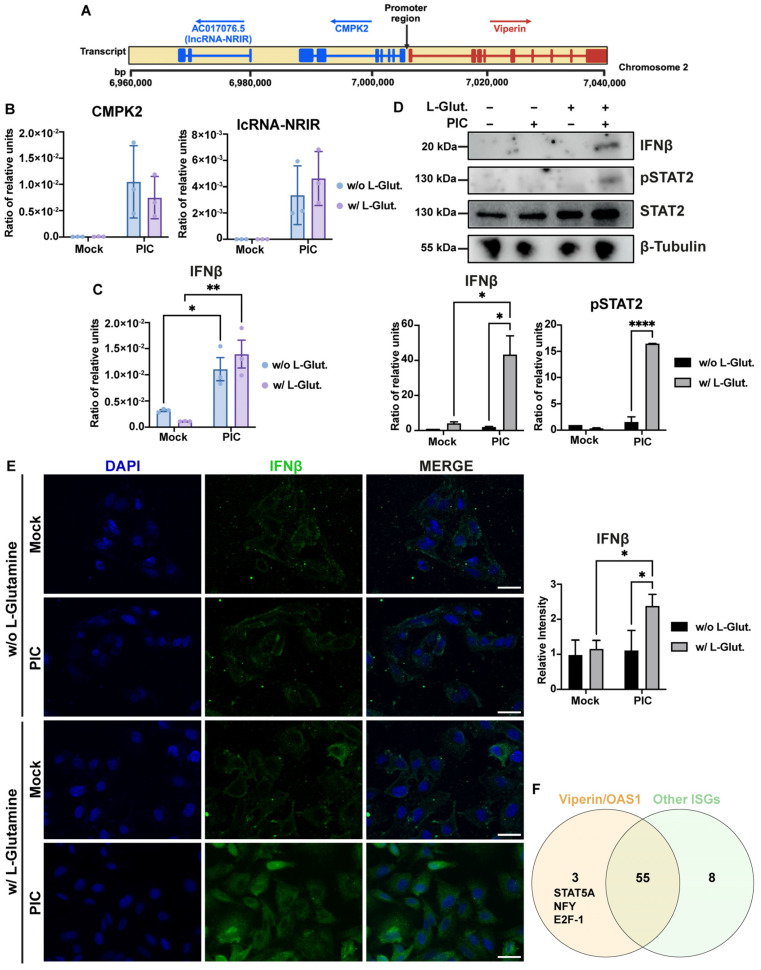
Differential expression of ISGs correlates with variable IFN-β signaling. (**A**) Schematic representation of viperin, CMPK2 and lc-NRIR loci on chromosome 2. Exon/intron structure of genes is represented, promoter region is annotated and arrows indicate orientation of transcription for each locus. (**B**) CMPK2 and lc-NRIR are expressed in same manner under glycolytic and OXPHOS conditions. Expression of CMPK2 and lc-NRIR has been assessed by RT-qPCR after 24 h stimulation of A549^Dual^ cells, grown in RPMI 2 g·L^−1^ glucose, supplemented or not with 2 mM L-glutamine, using poly:IC (PIC) 0.25 µg·mL^−1^. Results are expressed as mean normalized expression, following normalization with expression of RNA polymerase II RNA as housekeeping gene. (**C**) IFN-β mRNA has same level of expression under glycolytic and OXPHOS conditions. Expression of IFN-β has been assessed by RT-qPCR after 6 h stimulation of A549^Dual^ cells, grown in RPMI 2 g·L^−1^ glucose, supplemented or not with 2 mM L-glutamine, using PIC 0.25 µg·mL^−1^. Results are expressed as mean normalized expression, following normalization with expression of RNA polymerase II RNA as housekeeping gene. Statistical analyses were conducted using ANOVA with Dunnett’s correction. Error bars represent standard deviation of at least three independent experiments. * *p* < 0.0332; ** *p* < 0.0021. (**D**,**E**) Alternative ISGs expression is related to IFN-β production and signaling. IFN-β production and phosphorylation of STAT2 in response to PIC stimulation of A549^Dual^ cells grown in RPMI 2 g·L^−1^ glucose, supplemented or not with 2 mM L-glutamine, was assessed by western blot analysis and immunofluorescence. Western blot assay was achieved with mouse anti-IFN-β, rabbit anti-pSTAT2 and rabbit anti-β-tubulin antibodies. Immunofluorescence assay was achieved using mouse anti-IFN-β, donkey anti-mouse Alexa 488 and DAPI. These images are representative of three independent experiments. Scale bar: 10 µm. Statistical analyses were conducted using ANOVA with Dunnett’s correction. Error bars represent standard deviation of at least three independent experiments. * *p* < 0.0332; **** *p* < 0.0001. (**F**) Overexpression of viperin and OAS1 could be related to specific transcription factors activation under glycolytic condition. On one hand, we compared promoter region of viperin and OAS1, and on the other hand, promoter region of ISG12, ISG15 and ISG54, using PROMO tool based on TRANSFAC 8.3, looking for potential transcription factor binding sites. As represented, we determine three potential transcription factors that are specific for viperin and OAS1: STAT5A, NF-Y and E2F-1.

**Figure 7 viruses-16-01391-f007:**
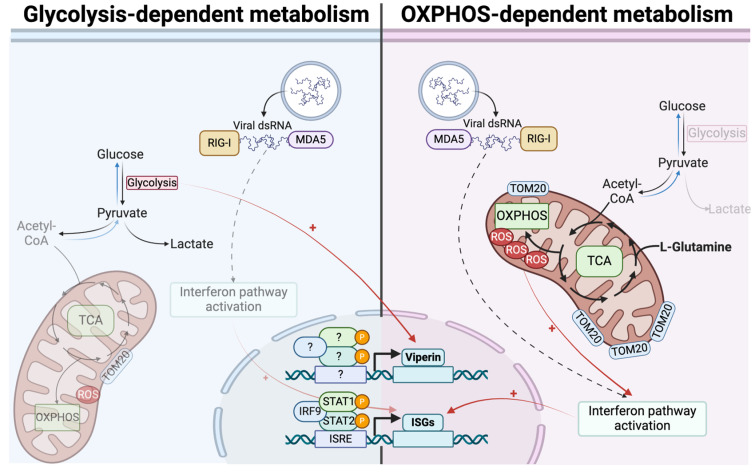
Bidirectional ISGs expression is based on differential metabolic states. In cells relying on OXPHOS, antiviral response mounting is related to more functional mitochondria, leading to higher expression of ISGs. On contrary, in cells relying on glycolysis, less efficient interferon signaling is associated with less functional mitochondria and consequently lower ISGs expression. Viperin and OAS1 escape this process and are more expressed in glycolysis-dependent cells, independently from interferon β.

**Table 1 viruses-16-01391-t001:** Primers sequences.

Gene	Forward Primer	Reverse Primer
RNA Polymerase II	5′-gagagcgttgagttccagaacc-3′	5′-tggatgtgtgcgttgctcagca-3′
18S ribosomal RNA	5′-agcctgcggcttaatttgac-3′	5′-caactaagaacggccatgca-3′
ISG56	5′-gcagccaagttttaccgaag-3′	5′-cacctcaaatgtgggctttt-3′
ISG54	5′-ctggtcacctggggaaacta-3′	5′-gagccttctcaaagcacacc-3′
ISG15	5′-agatcacccagaagatcggc-3′	5′-gaggttcgtcgcatttgtcc-3′
ISG12	5′-cgtcctccatagcagccaagat-3′	5′-acccaatggagcccaggatgaa-3′
IRF1	5′-gaggaggtgaaagaccagagca-3′	5′-tagcatctcggctggacttcga-3′
Viperin	5′-gatgttggtgtagaagaagc-3′	5′-ccaatccagcttcagatcag-3′
OAS1	5′-catgcaaatcaaccatgcca-3′	5′-acaaccaggtcagcgtcagatc-3′
CMPK2	5′-ccaggttgttgccatcgaag-3′	5′-caagagggtggtgactttaagag-3′
lncRNA-NRIR	5′-ccacccccacgaagaaattatatatc-3′	5′-gttagaggtgtctgctgcaataatc-3′
IFN-β	5′-cttggattcctacaaagaagcagc-3′	5′-tcctccttctggaactgctgca-3′

**Table 2 viruses-16-01391-t002:** Antibodies used for western blot.

Target	Reference	Dilution
Rabbit anti-ISG15	MA5-29371 (Invitrogen)	1:1000
Mouse anti-ISG56	MA5-25050 (Invitrogen)	1:1000
Mouse anti-viperin	MABF106 (Sigma-Aldrich)	1:1000
Mouse anti-IFN-β	AB_2122765 (Biolegend, San Diego, CA, USA)	1:1000
Rabbit anti-STAT2	72604 (Cell Signaling, Danvers, MA, USA)	1:1000
Rabbit anti-pSTAT2	07-224 (Sigma-Aldrich)	1:1000
Rabbit anti-β-tubulin	AC008 (ABclonal, Woburn, MA, USA)	1:1000
Mouse 4G2 (panflaviviral anti-E)	RD Biotech (Besançon, France)	1:1000
Goat anti-rabbit IgG-HRP	AB_2313567 (Jackson, West Grove, PA, USA)	1:5000
Goat anti-mouse IgG-HRP	AB_10015289 (Jackson)	1:5000

**Table 3 viruses-16-01391-t003:** Antibodies used for immunofluorescence.

Target	Reference	Dilution
Mouse anti-TOM20	sc-17764 (Santa Cruz, Dallas, TX, USA)	1:200
Mouse anti-IFN-β	AB_2122765 (Biolegend)	1:200
Donkey anti-mouse Alexa 488	ab150105 (abcam, Cambridge, UK)	1:750
Donkey anti-rabbit Alexa 488	ab150073 (abcam)	1:750

## Data Availability

All data needed to evaluate the conclusions in the paper are present in the article and/or the Appendix A. Additional data related to this paper are available from the corresponding author upon reasonable request.

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
