# Peer review of "Metabolic Dependency Shapes Bivalent Antiviral Response in Host Cells in Response to Poly:IC: The Role of Glutamine"

_viruses, 2024, doi:10.3390/v16091391_

Round 1

Reviewer 1 Report

Comments and Suggestions for Authors

The authors investigated the effect of various L-glutamine concentrations in cell culture medium on mitochondrial and cytoplasmic energy generation in a lung adenocarcinoma cell line A549 stably expressing luciferase under control of ISREs. They use an Oroboros Oxygraph to measure oxygen consumption as main metabolic read-out and luciferase activity and expression of selected ISGs as indicator of IFN induction by poly I:C. A key finding is that ISG54, ISG12 and ISG15 expression is increased under high L-glutamine concentrations (indicating that they are driven by oxidative phosphorylation) whereas expression of OAS1 was lower in the presence of L-glutamine. The latter was independently verified as depending on glycolysis. Thus, the IFN response to poly I:C stimulation (mimicking an RNA virus infection) has a bimodal metabolic dependency. The overall finding is of interest and current relevance to the fields of virology, immunology, and related fields. My main reservation is that the entire study was performed in a single cells type. A549 cells do reproduce many aspects of virus infection of primary cells, but it is well known that (as is typical of cancer cell lines), their metabolism does not entirely reflect that of nontransformed cells. I recommend that the authors repeat a key experiment in primary human respiratory cells. In support of the authors’ observations, however, one study performed in primary cells (in this case macrophages) comes to mind, and the authors may want to discuss their findings in the context of Waqas SF et al Clin Transl Med 2022  PMID 35842904, who report that chronic IFN signaling (due to ISG15 deletion) induces a preference for L-glutamine as carbon substrate.

Minor suggestion:  indicate the number of replicates (“n”) and the statistical test used in the legends.

Comments on the Quality of English Language

There are a few grammatical errors and typos. 

Reviewer 2 Report

Comments and Suggestions for Authors

The role of metabolism on the cell innate immune responses is a well-established concept and a topic of great importance that has attracted a high degree of scientific interest due to the implications in both basic and translational research in biology. However, the underlying mechanisms by which changes in cell metabolism impact the innate immune responses to infection remain poorly understood. Hence, the significance of the present work by Lebeau and colleagues examining the role of glutamine in the cell innate immune responses to viral infections, using poly:IC (pIC) as a surrogate of viral infection.

The authors have provided evidence of a bivalent response to pIC that depends on the metabolic state of the cell. They have convincingly shown that certain ISGs are expressed at higher levels under conditions favoring a metabolism based on oxidative phosphorylation, whereas other ISGs was enhanced in the absence of l-glutamine suggesting their dependence on a glycolytic cellular sate.  

The overall experimental design is sound, and the experimental section of the paper has been correctly done, including the appropriate controls. The description of the experiments contains sufficient details for their replicability by other investigators. The results presented support the key conclusions presented by the authors.

The central weakness of the paper relates on the lack of experiments showing a correlation between the metabolic induced changes in the expression of ISGs and protection, or lack off, against viral infection. The results shown in Fig. S2 did not adequately address this issue. It would be important to present results of DENV or ZIKV multi-step growth kinetics under oxidative and glycolytic metabolic conditions and correlate expression of ISGs with changes in viral growth.

Comments on the Quality of English Language

The overall quality of the writing is good, but some additional editing will help to improve the clarity of some of the sentences.

Reviewer 3 Report

Comments and Suggestions for Authors

Minor comments:

  1. Lines 307-314: It would be easier to follow if you labeled “Figure 1B - left panel,” “Figure 1B - middle panel,” and “Figure 1B - right panel” as Figure 1B, 1C, and 1D, respectively. Please change Figure 1 accordingly.

  2. Figure 1: To reduce the text in the figure, the two different conditions could be described as control (W/O L-Glut) and L-Glut (W/L-Glut). Otherwise, you need to abbreviate W/O & W/ in the figure legend.

  3. Figures 2 and 3: You need to define PIC in the figure legend.

  4. Figures 5 and 6: A few panel labels are missing. They should be labeled separately.

Round 2

Reviewer 1 Report

Comments and Suggestions for Authors

The authors have addressed my concerns adequately.

Reviewer 2 Report

Comments and Suggestions for Authors

In this revised version of their paper, the authors have responded to the main concern I raised during the review of the originally submitted paper by indicating that establishing a correlation between the cell metabolic state, ISG expression and viral growth is beyond the scope of the present work. In addition, the authors stated that they are currently unable to conduct experiments involving viral infections. Too address these issues the authors have added some content to the discussion of the revised version of their paper.

I appreciate the authors comments and the complications posed by not having access to facilities where viral infections can be done. 

I do not have any additional comments regarding the scientific content of this paper.

Comments on the Quality of English Language

The overall quality of the writing is fine.